# Object-Based Image Procedures for Assessing the Solar Energy Photovoltaic Potential of Heterogeneous Rooftops Using Airborne LiDAR and Orthophoto

**Arti Tiwari [1], Isaac A. Meir [2] and Arnon Karnieli [1],***

[1] The Remote Sensing Laboratory, Jacob Blaustein Institutes for Desert Research, Ben-Gurion University of the Negev, Beer-Sheva 85105, Israel; arti@post.bgu.ac.il

[2] Structural Engineering Dept., Faculty of Engineering Sciences, Ben-Gurion University of the Negev, Beer-Sheva 85105, Israel; sakis@bgu.ac.il

* Correspondence: karnieli@bgu.ac.il; Tel.: +972-52-8795925

**Abstract:** Available renewable energy resources play a vital role in fulfilling the energy demands of the increasing global population. To create a sustainable urban environment with the use of renewable energy in human habitats, a precise estimation of solar energy on building roofs is essential. The primary goal of this paper is to develop a procedure for measuring the rooftop solar energy photovoltaic potential over a heterogeneous urban environment that allows the estimation of solar energy yields on flat and pitched roof surfaces at different slopes and in different directions, along with multi-segment roofs on a single building. Because of the complex geometry of roofs, very high-resolution data, such as ortho-rectified aerial photography (orthophotos), and LiDAR data have been used to generate a new object-based algorithm to classify buildings. An overall accuracy index and a Kappa index of agreement (KIA) of 97.39% and 0.95, respectively, were achieved. The paper also develops a new model to create an aspect-slope map, which combines slope orientation with the gradient of the slope and uses it to demonstrate the collective results. This study allows the assessment of solar energy yields through defining solar irradiances in units of pixels over a specific time period. It might be beneficial in terms of more efficient measurements for solar panel installations and more accurate calculations of solar radiation for residents and commercial energy investors.

**Keywords:** object-based image analysis; segmentation; LiDAR; orthophoto; aspect-slope map; solar energy photovoltaic potential

## 1. Introduction

Global energy demand is gradually increasing every day, and most consumption still relies on fossil fuels [1]. This results in the rapidly growing overutilization of natural resources and the emission of anthropogenic greenhouse gases (GHGs). The critical issues of anthropogenic climate change, the energy crunch, and the environmental pollution produced by the combustion of fossil fuels are major concerns worldwide [2]. It is widely accepted that society needs a change in energy production and consumption [3]. Consequently, it is necessary to replace fossil fuels with renewable energies (i.e., solar, wind, hydropower, etc.) [2].

Renewable energy is considered to be the best solution for various energy challenges, for example, the exhaustion of fossil fuels and environmental pollution, since it significantly contributes to environmental protection, commercial growth, and energy safety [4]. Thus, in the quest for sustainable development, significant efforts have been invested in exploiting renewable energies, with local governments offering incentives for decreasing conventional energy expenses. Among all the available

renewable energy sources, solar energy dominates over others in terms of its energy capacity growth rate and low maintenance cost [5]. Subsequently, solar energy has been accepted as a clean, abundant, readily available, free, environmentally friendly, and economic energy source [6], without carbon cost [7]. The installation of photovoltaic panels on the building roofs has various advantages such as efficient utilization of renewable solar energy and distribution of total residential energy consumption that eventually leads to the reduction of $CO_2$ emission [8]. In urban areas, building roofs are considered to be more appropriate than building facades as spaces on which to install solar panels [9,10]. In addition, solar photovoltaic (PV) technologies offer the most cost-effective investment, partially since such systems need little maintenance to produce electricity free of GHGs [11]. Renewable solar energy helps in reducing the emission of GHGs, which also supports green building ranking schemes [12]. Thus, solar energy contributes to buildings' energy requirements and provides health benefits, environmental stewardship, and more sustainable cities. Earlier related research aimed at identifying and characterizing the reduced manufacturing costs of photovoltaic devices [13].

Solar energy potential is an important attribute in identifying the most appropriate building roof surfaces on which to mount solar panels [14]. The efficient utilization of solar energy and fulfillment of energy demand for both commercial and residential buildings is still a challenge. Therefore, it is important to accurately estimate solar energy potential and examine its distribution over building roofs. This is not simple as it depends on considerable spatial and temporal variations in solar radiation that are significantly affected by several factors [15], including different roof geometries [16], inclined surfaces [17], shadow effects [18], geographical locations [19], and topographic features [20]. Thus far, several models and technologies have been studied to estimate the energy potential of building roofs [21].

Light detection and ranging (LiDAR) mapping is one of these techniques [22], primarily since the elevation data in urban surroundings can be easily attained through airborne LiDAR scans. Secondly, LiDAR-based data acquisition technology permits the quick reorganization of terrain surfaces when estimating building insolation with existing two-dimensional models. Using 2.5D DSM, Carneiro et al. [23] established a tool to estimate solar energy that is not only appropriate for building roofs but also suitable for the study of facades. Redweik et al. [24] then suggested a method to estimate solar energy based on the r.sun radiation model, established by Šúri et al. [25], and integrated this into the open-source Geographic Resources Analysis Support System (GRASS-GIS) [26]. Despite having larger areas, the result showed that the building facades return lower solar energy than roofs. A method has been presented to estimate solar potential by using graphic processing units (GPU) with measured unified device architecture technology and a LiDAR dataset [19]. The LiDAR methods significantly improve the energy potential estimation in terms of the anticipated model error [27], and they have been frequently used in the calculation of solar energy potential in urban settings by incorporating GIS technologies [28,29]. Though LiDAR-based techniques are cost-effective and suitable for buildings, there are many restrictions in terms of computation cost and data availability, and they are limited to the comparatively simple structure of building roofs. In order to overcome such limitations, new methodologies are essential for estimating solar energy. A pixel-based method was applied to estimate solar energy on flat roofs [30], while Li and Liu [31] applied a pixel-based approach to pitched roofs.

The primary aim of the current study is to develop a procedure for assessing the rooftop solar energy photovoltaic potential over a heterogeneous urban environment. Such an environment may include flat and pitched roof surfaces at different inclinations and in different directions, along with multi-segment roofs on a single building. Due to the complex roof geometry, very high-resolution data, such as ortho-rectified aerial photography (orthophotos), along with LiDAR data, may be an appropriate means for accomplishing the goal. The previously reported studies show that the aspect and slope information plays a very crucial role in solar estimation analysis [27,32–34]. Therefore, detailed aspect and slope maps are essential. Using the products of airborne LiDAR, a new model was developed to create an aspect-slope map. A minor change in slope and its direction over rooftops is noticeable while working on a single map (i.e., aspect-slope map) that could be overlooked while



observing individual maps. Although various studies have been reported to estimate solar energy over building roofs but more specifically, to get the building footprints, an object-based method was applied to segment building as an object [32]. In some studies, building polygons was directly considered to estimate the PV potential [34–36]. To automate this tedious task, the present study develops an object-based algorithm to classify all buildings of the study area. Furthermore, the research assumes that in order to accurately obtain the value of the roof's aspect and slope, an object-based method, using a data fusion technique, should be applied.

In the paper, Section 2, describes the complete methodology which is implemented to estimate the PV potential on the building rooftops. The section goes through the various steps including study area and data sources information, applied object-based classification algorithm, LiDAR generated DSM used to produce an aspect-slope map, and solar radiation maps. Finally, spatial analysis has performed over building roofs. In Section 3, results and discussion has been performed. The subsections of Section 3, contain the classification and accuracy assessment, outcome of LiDAR mapping, validation of generated aspect-slope map, spatio-temporal distribution of solar radiation, and final results of spatial analysis. Finally, the work has been summarized in Section 4.

## 2. Methodology

### 2.1. Study Area and Data Sources

The developed methodology was demonstrated in a neighborhood in the small city of Kiryat Malakhi, located in the southern coastal plain of Israel (31°44′N 34°44′E) (Figure 1). This particular neighborhood, ~0.45 km² in size, was selected since it contains tens of buildings, one to three stories, with various types of roofs, flat and pitched, in different inclinations and orientations, along with multi-segment shapes.

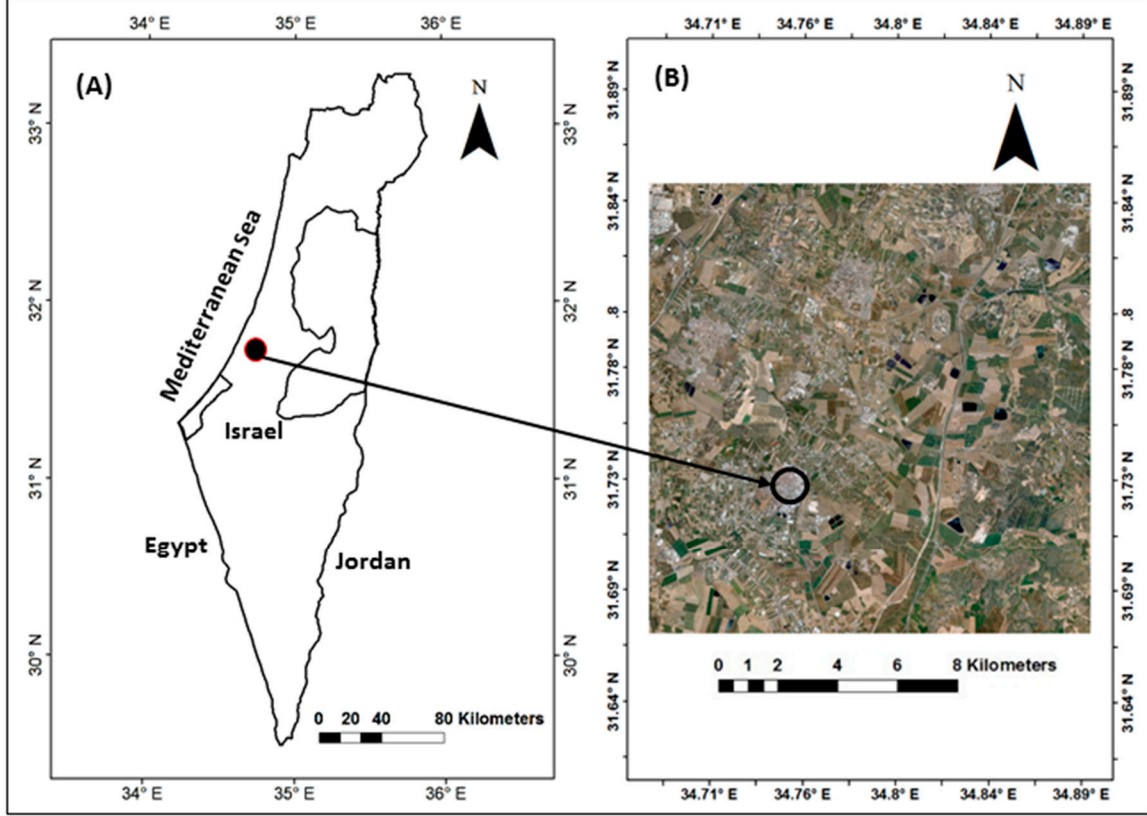

**Figure 1.** (**A**) Location of the study area in Israel, (**B**) zoom-in on the city of Kiryat Malakhi.

Three different data sources were used (Figure 2). The first is an orthophoto, a rectified aerial photograph that was obtained in 2012 at a geometric resolution of 25 cm per pixel, purchased from the Survey of Israel. It contains standard red-green-blue (RGB) bands with eight-bit radiometric resolution. The other two are a digital terrain model (DTM) and a digital surface model (DSM), both created by an airborne light detection and ranging (LiDAR) map in 2015. The DTM represents the elevation of the ground, while the DSM reflects the height of the elevated structures at each point, e.g., buildings. The DTM was created with a 1-m pixel resolution by the least elevation and referred to as the "last return" in LiDAR data processing terms, whereas the DSM with the highest elevation in the same pixel was referred to as the "first return". The study assumes minimal surface changes between 2012 and 2015.

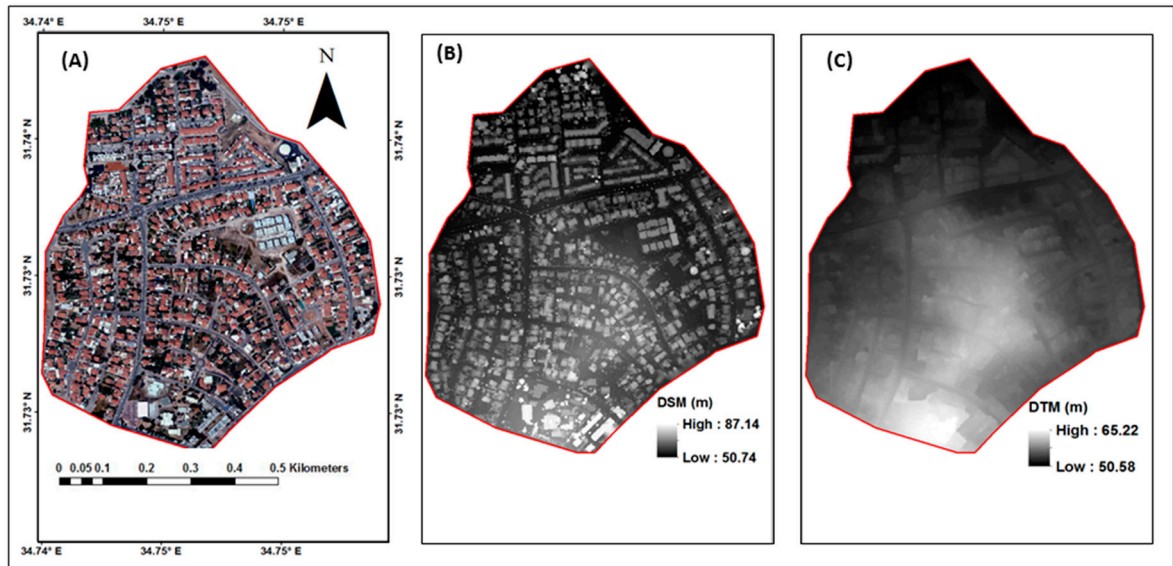

**Figure 2.** (**A**) A true-color orthophoto image of the study area, (**B**) digital surface model (DSM), (**C**) digital terrain model (DTM).

### 2.2. Classification

An object-based classification algorithm was applied to extract all buildings in the study area using eCognition Developer 9.3. The initial step in the object-based approach is to produce segmented objects that contain a cluster of pixels with uniform feature values. The current work applied a multi-level segmentation hierarchy, i.e., Level-I and Level-II (Figure 3), in order to optimize the image segments. The Level-I multiresolution segmentation was achieved by combining the orthophoto and the LiDAR-derived DSM and DTM raster layers. Table 1 lists the used parameters for generating the segmented layer of the multiresolution algorithm. The algorithm is a bottom-up region-merging approach starting with the one-pixel object. It selects the initial pixel randomly in the image, and then depending on the weighted heterogeneity factors, i.e., segment size and heterogeneity, it chooses whether the adjacent objects should be combined or not. The neighboring objects with less heterogeneity are fused [36].

**Table 1.** Parameters of the multiresolution segmentation algorithm.

| Domain | Image Layer Weights | Scale Parameter | Shape Factor | Compactness |
|---|---|---|---|---|
| Pixel level | R, G, B, DTM = 1, DSM = 3 | 10 | 0.5 | 0.7 |

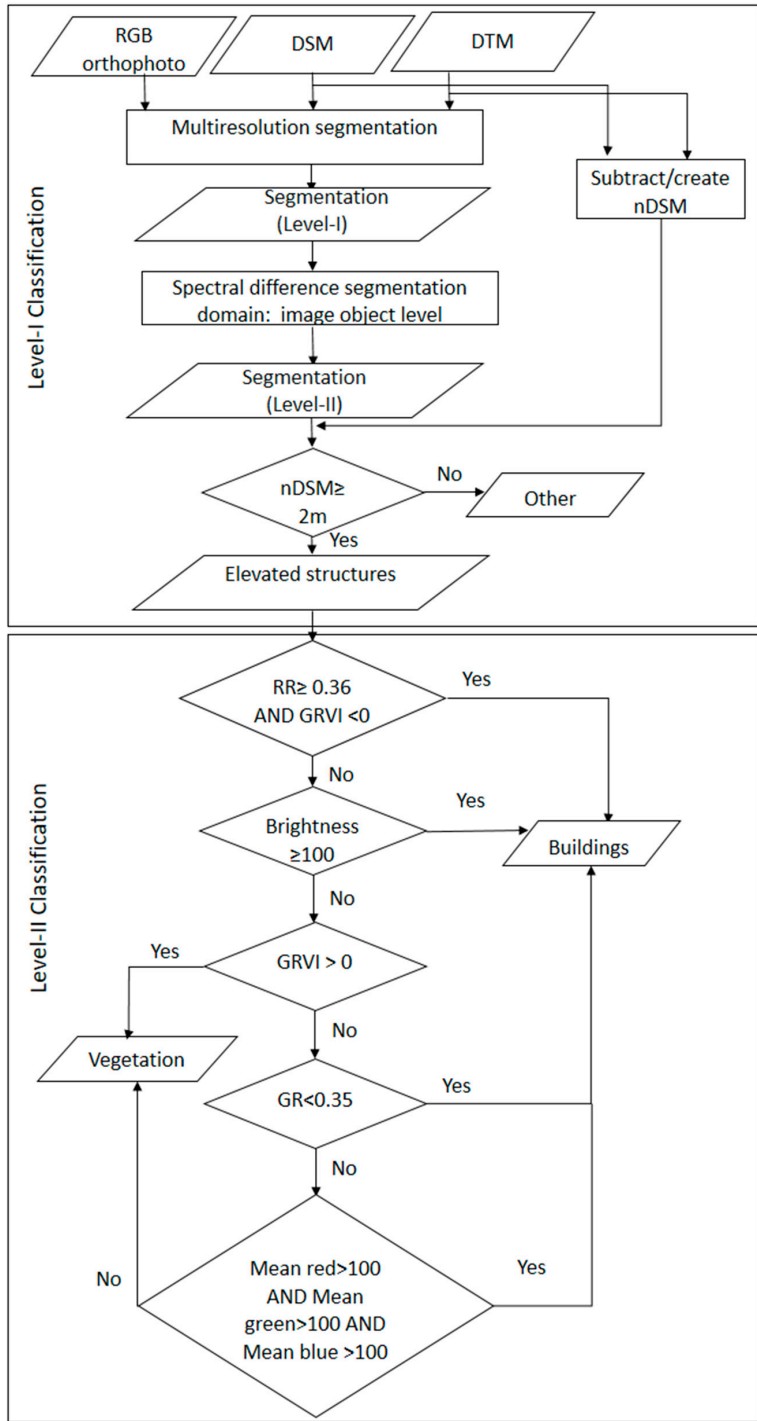

**Figure 3.** Created image hierarchies: Segmentation Level-I and Level-II and the implemented methodology to achieve Level-I and Level-II classification.

Spectral difference segmentation was applied to achieve sub-level objects of image hierarchy, i.e., Level-II. The algorithm combines neighboring image objects, based on their mean image layer intensity values. Merging of neighboring image objects occurs if the difference between the layer mean intensities is lower than the value specified by the maximum spectral difference (five, in the current case). This algorithm is aimed at improving the current segmentation results by combining spectrally similar image objects created by prior segmentations. The algorithm cannot be used to produce new image object levels depending on the pixel level domain [36].

To classify buildings, an object-based image classification algorithm was applied to the segmented image hierarchy Level-II along with a normalized digital surface model (nDSM) (Figure 3). nDSM is the difference between the DSM and DTM heights, and thus represents the height of the elevated structures. All objects with an nDSM ≥ 2 m were assigned to the "elevated structures" class, and the remaining features were named as "other" to receive the Level-I classification. Since some elevated structures also include vegetation, a few additional features were customized to improve the Level-II classification (Figure 3). To enhance the spectral difference between bands, ratio images were effectively used in mapping varied zones [37]. Two ratio images, the red ratio (RR) and the green ratio (GR), were employed to identify vegetation within the elevated area. In addition, the green-red vegetation index (GRVI) was calculated using the RGB orthophoto bands. Motohka et al. [38] analyzed the GRVI to monitor seasonal change with time. The GRVI is derived as:

$$GRVI = \left(\rho_{green} - \rho_{red}\right)/\left(\rho_{green} + \rho_{red}\right) \tag{1}$$

where $\rho$ is reflectance at a particular spectral band.

Other parameters, such as the brightness and mean values of each RGB band, were incorporated to increase the classification accuracy. To quantify this procedure, the following algorithms were established, ending with two classes, namely buildings and vegetation:

$$\text{Buildings} \quad (RR \geq 0.36 \wedge GRVI < 0) and (Brightness \geq 100) \ and (GR < 0.35) \ and \ (Mean \ R > 100 \wedge Mean \quad G > 100 \ \wedge Mean \ B > 100) \tag{2}$$

$$\text{Vegetation} \quad (GRVI > 0) \tag{3}$$

The classification accuracy was evaluated with generated reference values based on acquired samples using a very high-resolution orthophoto. The selected samples were compared to the classification product. The overall accuracy and Kappa coefficient were obtained by using an error confusion matrix [39,40].

### 2.3. Aspect and Slope

The DSM values were applied to execute the aspect tools present in the ArcGIS toolbox. Aspect maps provide the observer with a sense of each slope's orientation. It was achieved by separating the angle of the slope into eight quadrants: North (0°), Northwest, West (270°), Southwest, South (180°), Southeast, East (90°), and Northeast. If a given surface is flat, the aspect value was assigned as −1. The aspect was measured in degrees in the clockwise direction from north 0–360°.

Slopes are important in terms of how steep a surface is. Usually, this can be measured as either a degree or percent/ratio value from one raster cell to the following one of the DSM. In the slope map, 0-degree corresponds to a flat surface, while a 90-degree value specifies a completely vertical surface. The DSM was used to generate a slope map using the slope tool in ArcGIS.

In order to produce an aspect-slope map, the two data sets should be incorporated into a single one. This particular map merges slope orientation (aspect) with the gradient of the slope (in degrees) and uses color and saturation together to demonstrate the collective results. To generate such a map, a model was developed in ArcGIS based on the DSM raster file. The original aspect values were reclassified into 10 to 80 new class values (Table 2) and the slope values in the range of 0 to 9 (Table 3). To produce a single aspect-slope raster grid, the new slope and aspect values were joined together. In the combined raster grid, pixel values range anywhere from 10 to 89, which is the minimum aspect value plus the minimum slope value (10 + 0, i.e., 10) and the maximum aspect value plus the maximum slope value (80 + 9, i.e., 89). In the generated aspect-slope map, the "tens column" of the pixel value represents the direction of slope, i.e., aspect, and the "unit column" represents slope.

**Table 2.** Original aspect values (degrees) and assigned re-class values.

| Original Aspect (in Degrees) Value | Re-Class Value |
|---|---|
| ≥0 and ≤22.5 | 10 |
| >22.5 and ≤67.5 | 20 |
| >67.5 and ≤112.5 | 30 |
| >112.5 and ≤157.5 | 40 |
| >157.5 and ≤202.5 | 50 |
| >202.5 and ≤247.5 | 60 |
| >247.5 and ≤292.5 | 70 |
| >292.5 and ≤337.5 | 80 |
| >337.5 and ≤360 | 10 |

**Table 3.** Original slope values (%) and assigned re-class values.

| Original Slope (%) Value | Re-Class Value |
|---|---|
| ≥0.0 and ≤23.05 | 0 |
| >23.05 and ≤65.30 | 1 |
| >65.30 and ≤115.23 | 2 |
| >115.23 and ≤169.00 | 3 |
| >169.00 and ≤226.61 | 4 |
| >226.61 and ≤288.07 | 5 |
| >288.07 and ≤357.20 | 6 |
| >357.20 and ≤453.23 | 7 |
| >453.23 and ≤587.66 | 8 |
| >587.66 and ≤979.43 | 9 |

In the study area, pitched roofs were more difficult to analyze within the generated data of the aspect-slope map. Though the weighted slope for all slope classes was calculated and based on the value, the slope was converted to degrees for further calculation. The calculated weighted slope would determine how much impact each slope would have on the final results. To fulfill the goal of the study, the slopes were grouped into four major categories (Table 4): Flat (<4°), gentle slope (4–22°), moderate slope (22–40°), and steep slope (>40°).

**Table 4.** Categorization of slope.

| Original Slope (%) Value | Re-Class Value | Slope in Degrees | Categorized Slope in Degrees | Description |
|---|---|---|---|---|
| ≥0.0 and ≤23.05 | 0 | 3.05 | <4° | Flat |
| >23.05 and ≤65.30 | 1 | 8.59 | 4–9° | Gentle slope |
| >65.30 and ≤115.23 | 2 | 14.93 | 9–15° | |
| >115.23 and ≤169.00 | 3 | 21.36 | 15–22° | |
| >169.00 and ≤226.61 | 4 | 27.67 | 22–28° | Moderate slope |
| >226.61 and ≤288.07 | 5 | 33.68 | 28–34° | |
| >288.07 and ≤357.20 | 6 | 39.57 | 34–40° | |
| >357.20 and ≤453.23 | 7 | 46.36 | 40–47° | Steep slope |
| >453.23 and ≤587.66 | 8 | 53.67 | 47–54° | |
| >587.66 and ≤979.43 | 9 | 66.19 | 54–67° | |

## 2.4. Solar Radiation

Estimating solar radiation using GIS is an effective way to produce an insolation map and link it to additional geospatial data [41]. The Esri's Solar Analyst tool was developed for precise mapping on point-specific radiation models, alongside prompt and accurate measurement of solar radiation maps of a heterogeneous urban environment [42]. The Solar Analyst tool provides accurate measurements of solar insolation using a GIS-based hemispherical viewshed model, initially developed by Rich [43] and later improved by Fu and Rich [42]. The algorithm measures the sky obstruction and incoming radiation for each raster grid for the given digital elevation model (DEM), as the visible sky depends on landscape topography [42].

The estimated solar radiation is attained by adding the direct and diffuse radiation of a specific point or a particular area, and the summation of these determines the total global radiation in Watt-hours per meter squared (Wh/m$^2$) units. To calculate direct and diffuse solar radiation, the measured viewshed is overlapped on the direct sun map and diffuse sky map, respectively [44]. The sun map is a two-dimensional representation that detects sun tracks, marking the apparent position of the sun as it varies with time for the identical location. To generate sky maps, the entire sky is divided into a sequence of sky sectors demarcated by zenith and azimuth divisions. Sky segments in the sky map must be small enough so that the centroid zenith and azimuth angles reasonably signify the direction of sky sectors in the resulting estimations.

The Solar Analyst tool has some limitations in its execution, as it excludes reflected radiation in the measurement. However, Fu and Rich [42] claim that the influence of reflected radiation is usually very small except in high albedo areas. Additionally, the tool does not deal with cloud cover directly. Therefore, the tool's accuracy depends on the applied input digital elevation models. If the environment and the topography can significantly affect the measured solar radiation for the given surface, then accurate measurements can be very challenging [42].

The Esri's area solar radiation tool was the preferable GIS modeling tool for calculating solar radiation for each cell in a raster image based on topographic features and geographic location of the area. The study focused on specific days of the year, i.e., the vernal equinox (21 March), autumnal equinox (23 September), winter solstice (22 December), and summer solstice (21 June), because of the very high transition in-season change or sun position. Solar energy radiation distribution from the vernal equinox to the summer solstice and from the autumnal equinox to the winter solstice can be better envisaged, throughout the year, on the basis of instantaneous incoming solar insolation. The primary input for the tool was the LiDAR-derived DSM that provided the latitude, as well as the aspect and slope value for each pixel. The three time periods (i.e., 9:00, 12:00, and 15:00) were considered for the solar insolation calculation because the maximum fluctuation of solar radiation can be observed in the mentioned time frames.

## 2.5. Spatial Analysis of Building Rooftops

Spatial analysis was performed over building roofs for a better estimation of PV potential. A functional methodology of the performed analysis is described in Figure 4. In object-based classification, the building class was masked out as a polygon vector layer to perform further analysis that included the height of identified buildings. The solar radiation raster layer and aspect-slope map of the above-mentioned four days of the year were clipped according to the building class. Once the roofs were arranged together with all spatial data, it was possible to incorporate the results for calculating PV potential on an individual rooftop. In order to perform spatial analysis with building rooftop polygons, vector point layers were generated for all raster images (i.e., aspect-slope and solar radiation). Employing the raster to point tool, an individual pixel was converted into a point vector feature. All pixel values of aspect, slope, elevation, and solar radiation raster became an attribute of the point vector layer feature class. After preparing all attribute files, a single feature class layer was generated using spatial join. The feature class layer contains all information on elevation, aspect, slope, and solar radiation value at each point over a building rooftop. Thus, this spatial join made

it feasible to determine the actual slope and aspect of building roofs that provide the maximum solar potential. Firstly, the solar radiation estimation was measured aspect-wise, and later slope categorization took place.

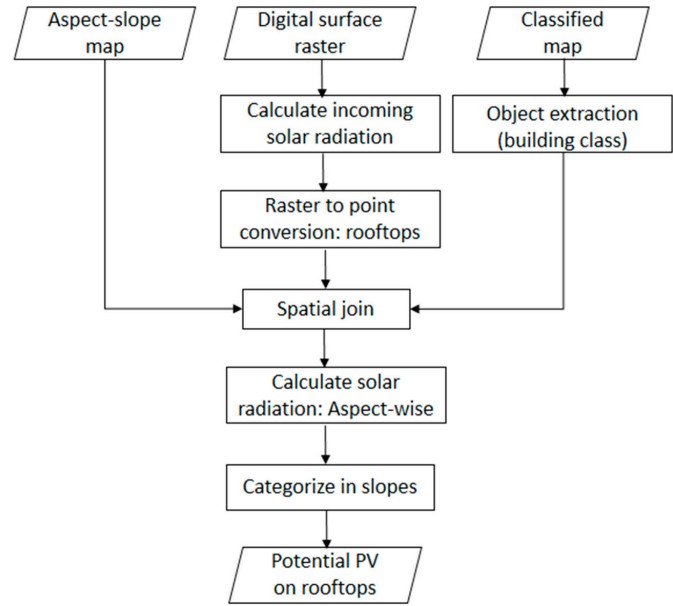

**Figure 4.** The implemented methodology to estimate photovoltaic (PV) potential.

## 3. Results and Discussion

### 3.1. Classification and Accuracy Assessment

Although various studies have estimated solar energy on building roofs, to obtain the building footprints specifically, here, an on-screen digitization process was used to generate building polygons. To address this tedious task, an object-based algorithm was developed to extract all buildings in the studied area.

The method of classification was divided into two levels. Level-I classification output yielded a general sight of elevated areas in the study area (Figure 5A), and Level-II classification subdivided the image objects into two classes, namely buildings and vegetation (Figure 5B).

For accuracy assessment, the selected reference samples were matched with the final classification, and numerical measurements were performed using the eCognition software. The classification resulted in an overall accuracy of 97.39% based on a randomly selected individual sample reference, though there was some misclassification between the mentioned classes. The user and producer accuracies of the building classes are 99.38% and 98.03%, respectively. The Kappa index of agreement is 0.95. (Table 5).

**Table 5.** Confusion matrix for object-based image classification.

| Classification/Reference Class | Buildings | Vegetation | Sum | User Accuracy (%) |
|---|---|---|---|---|
| Buildings | 52,720 | 328 | 53,048 | 99.38 |
| Vegetation | 537 | 47,651 | 48,188 | 98.89 |
| Unclassified | 523 | 1303 | 1826 | |
| Sum | 53,780 | 49,282 | | |
| Producer Accuracy (%) | 98.03 | 96.69 | | |
| Overall Accuracy 97.39 | | | | |
| Kappa index of agreement 0.95 | | | | |

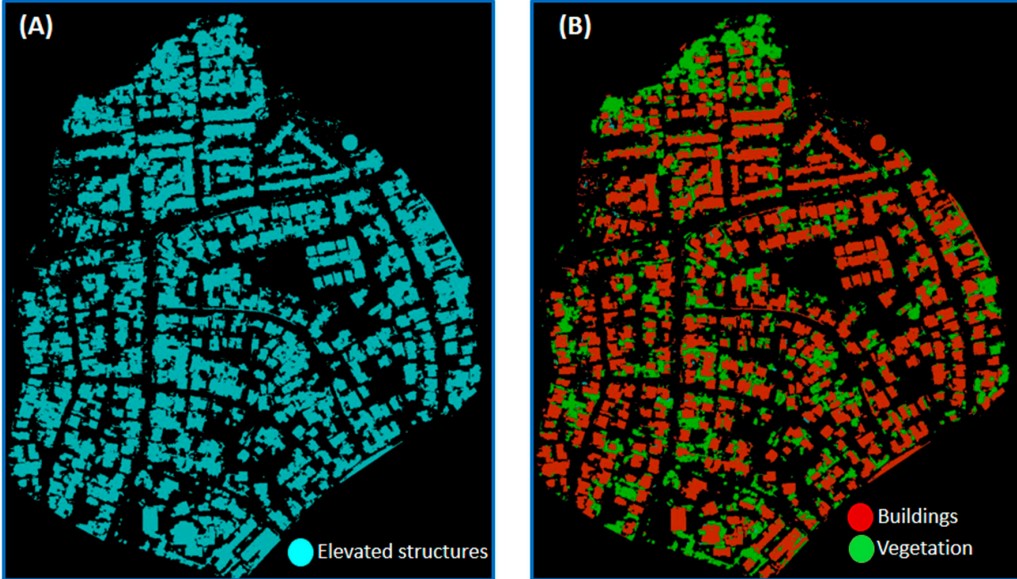

**Figure 5.** (**A**) Classification of all elevated structures, (**B**) separation of buildings and vegetation.

### 3.2. LiDAR Mapping

The LiDAR-generated aspect and slope maps, based on their reclassifications, are presented in Figure 6A,B. The analysis contributes to a better understanding of several different aspects and their respective slopes, which can affect a single rooftop's solar potential.

Incorporating these two maps into a single one produces the aspect-slope map, shown in Figure 7. Different slope directions are marked in different colors, while the slope steepness is indicated by the brightness of the color (steeper slopes are brighter in color). Note that the area has a pitched roof and, in places, multiple roof segments with different slopes and directions on a single building roof. The aspect-slope frequency distribution values of the rooftop segments are presented in Figure 8, showing that 6%–10% of the area is categorized as gentle slopes (4–22°) with changing aspects. Flat, moderate, and steep categories also exist but are less numerous.

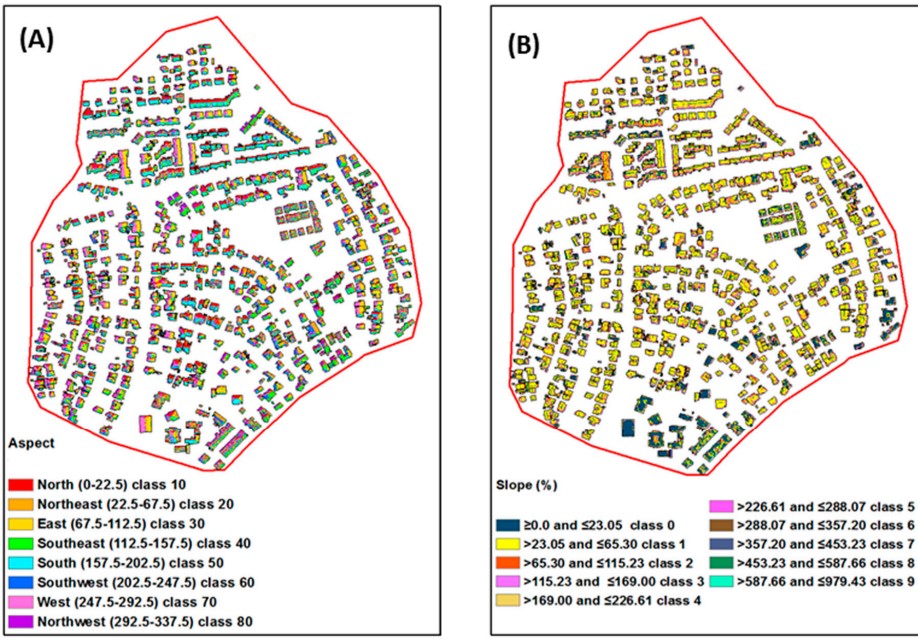

**Figure 6.** (**A**) Aspect reclassification, (**B**) slope reclassification.

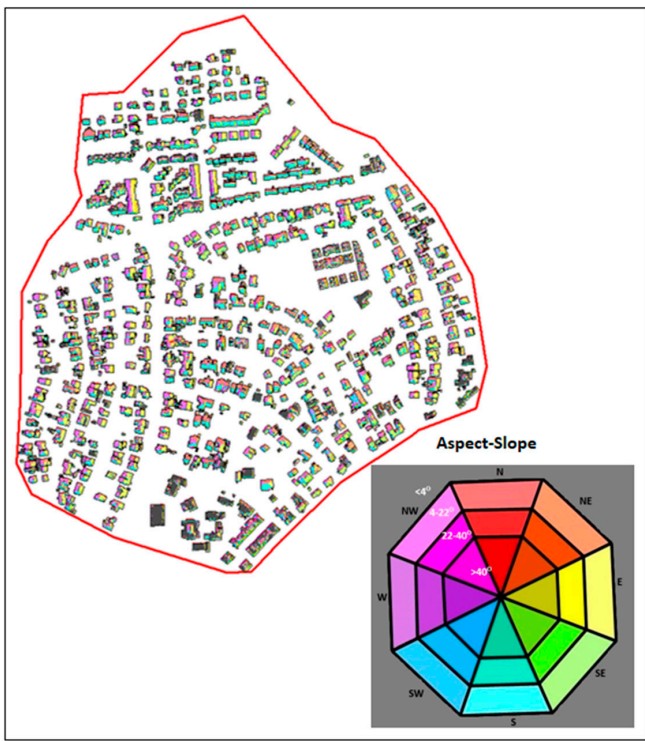

**Figure 7.** An aspect-slope map.

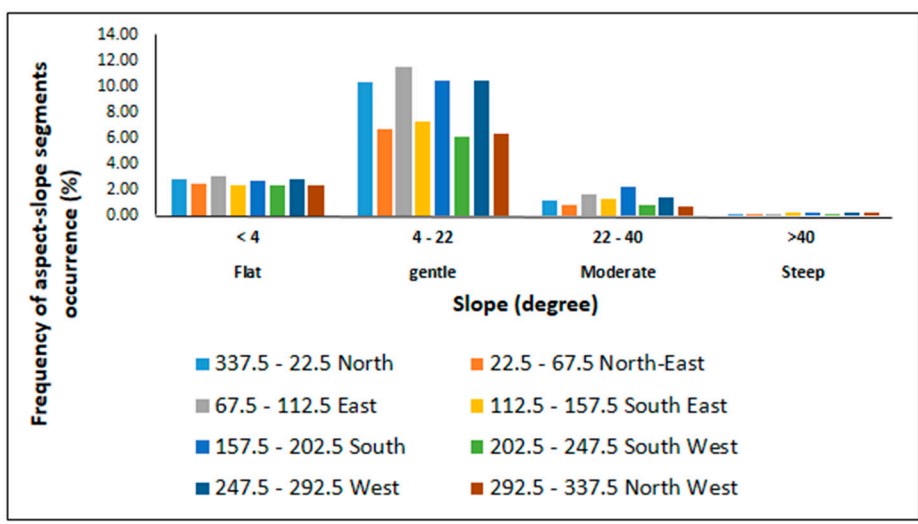

**Figure 8.** Frequency distribution of aspect-slope rooftop segments.

### 3.3. Visual Validation of Aspect-Slope Map

Since accurate measurements of the roof orientations and slopes in the study area were technically impossible, validation of the aspect-slope map was performed by ground visual examination of the building. A large number of buildings were observed from the street level to verify the LiDAR-derived map. It is assumed that no significant land cover change had occurred between the dates of orthophoto and LiDAR data collection, except for very minor structure modifications that accrued in recent years. Panels 1, 2, and 3 in Figure 9 demonstrates flat roofs. Gentle slope (4–22°) with north-south orientation are presented in panels 4, and 5, while east-west orientation shows in panels 6 and 7. Panels 8, 9, and 10 demonstrate multi-segment roofs with gentle slopes. Few pixels over a multi-segment and sloped roof correspond to the moderate (22–40°) and steep slope (>40°) categories in the generated

aspect-slope map. It was a noticeable point during the field campaign that the pixels correspond to the steep slope (>40°) present at the edges of the roofs.

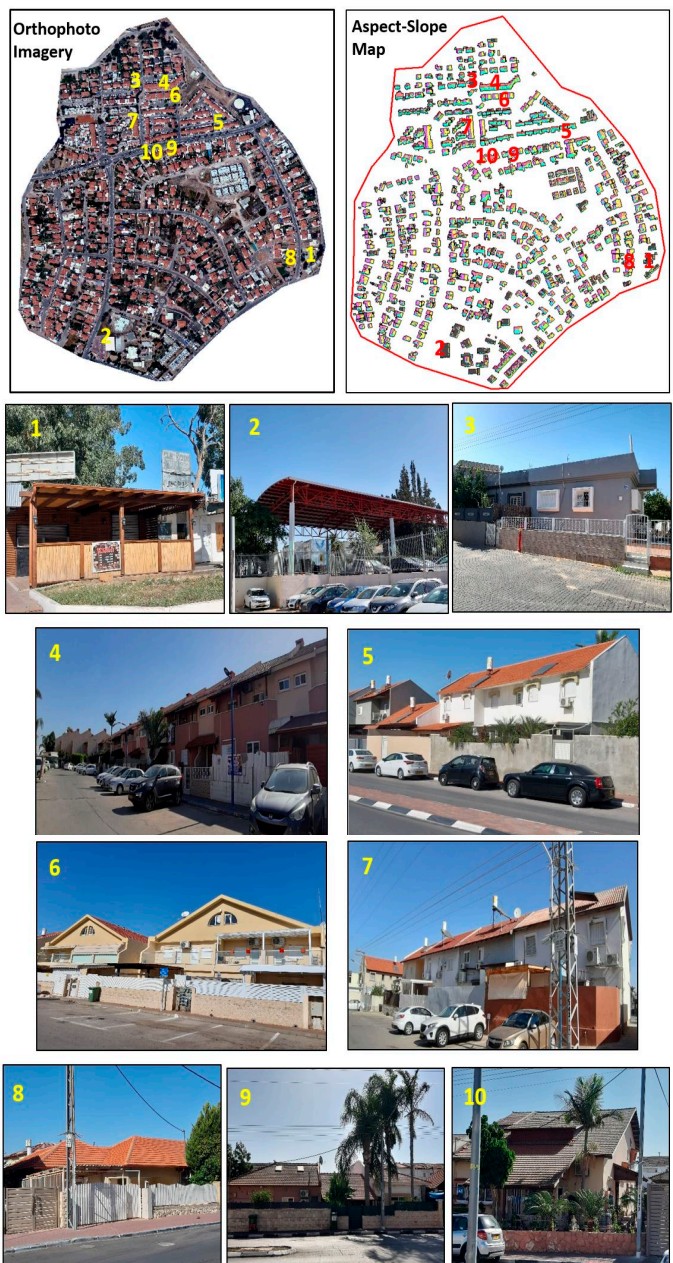

**Figure 9.** Validation of the aspect-slope map: **1**, **2**, **3**: Flat roof (<4°), **4**, **5**: North-South (N-S) with gentle slope (4–22°), **6**, **7**: East-West (E-W) with gentle slope (4–22°), **8**, **9**, **10**: multi-segment roof with different directions.

### 3.4. Spatio-Temporal Distribution of Solar Radiation

Solar radiation maps are still the most commonly used method to demonstrate the geographic variation and intensity of solar insolation on rooftops. By employing the Esri ArcGIS Area Solar Radiation tool and the LiDAR-generated DSM, direct and diffuse solar radiation values, as well as instantaneous solar insolation for individual raster grid cells, were computed for 9:00, 12:00, and 15:00 for specific days of the year, i.e., the vernal equinox (21 March), autumnal equinox (23 September), winter solstice (22 December), and summer solstice (21 June). In the generated solar radiation maps, the values were scaled from "0 to 23" for better visualization of the differences between the specific

days of the year. Figure 10 shows the spatio-temporal distribution of the total incoming solar radiation (direct and diffuse) for the study area at 12:00 for all four days of the year. In this figure, minimum incoming solar radiation values are presented in blue and maximum values in red. In the generated maps of Figures 10–12, 'A' represents the vernal equinox (21 March), 'B' the autumnal equinox (23 September), 'C' the winter solstice (22 December), and 'D' the summer solstice (21 June). Further detailed maps at 9:00 and 15:00 are included as Figures 11 and 12, respectively. Figures 10–12 reveal that the winter solstice (22 December) and summer solstice (21 June) received the minimum and maximum solar radiation, respectively. The Equinox days, however, have minor changes in radiation values.

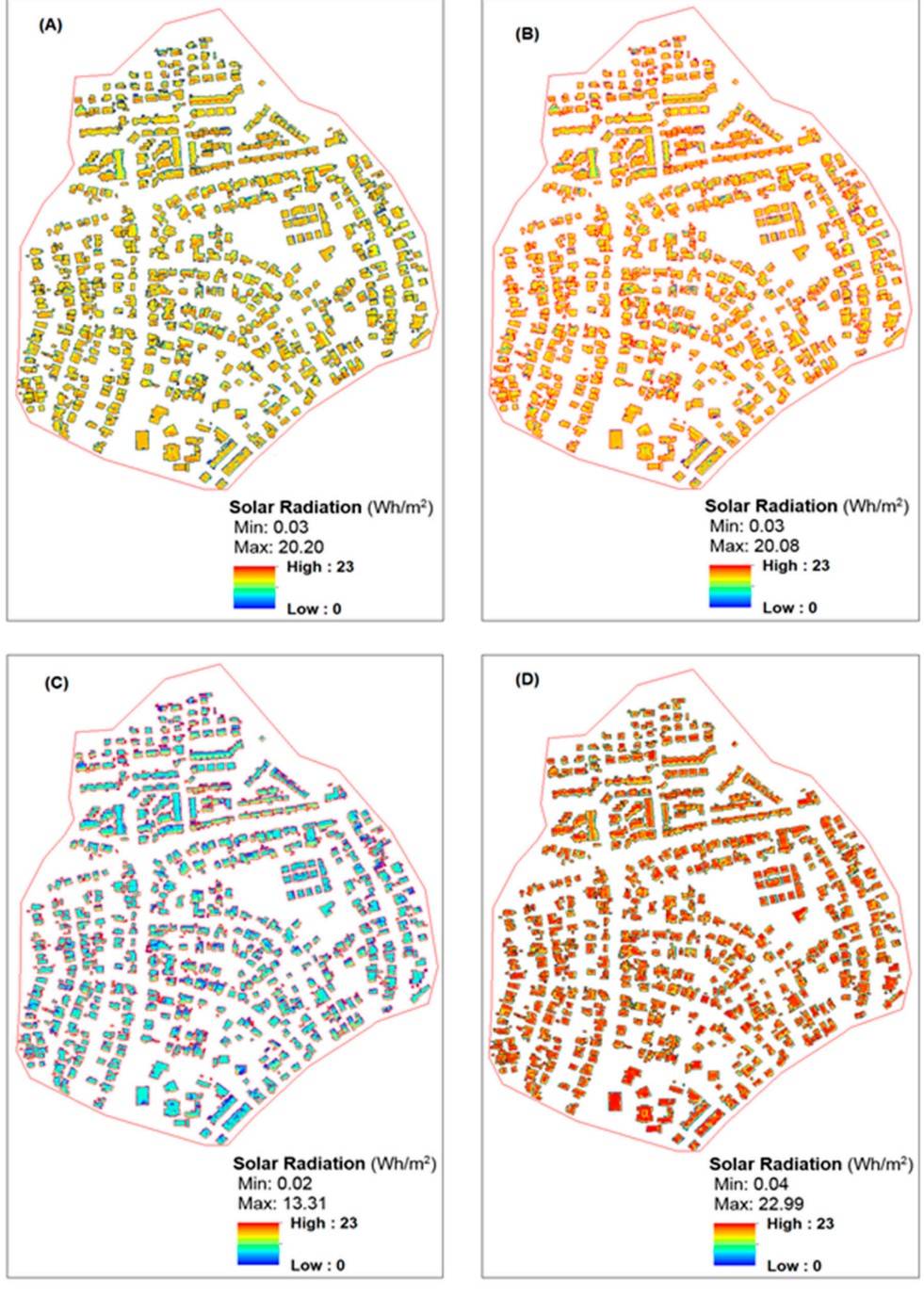

**Figure 10.** Incoming solar radiation (Wh/m$^2$) at 12:00: (**A**) vernal equinox (21 March), (**B**) autumnal equinox (23 September), (**C**) winter solstice (22 December), and (**D**) summer solstice (21 June).

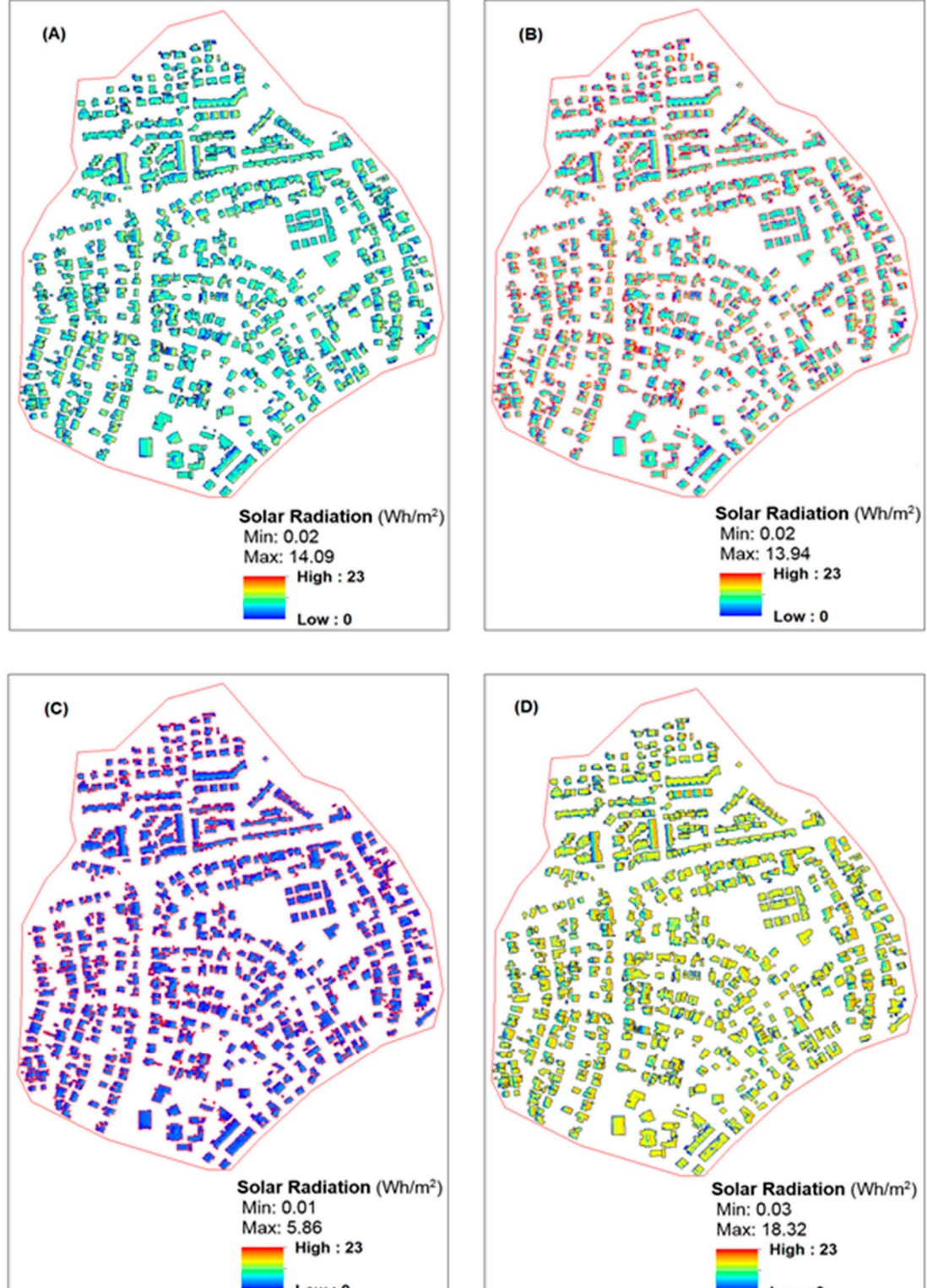

**Figure 11.** Incoming solar radiation (Wh/m$^2$) at 9:00: (**A**) vernal equinox (21 March), (**B**) autumnal equinox (23 September), (**C**) winter solstice (22 December), and (**D**) summer solstice (21 June).

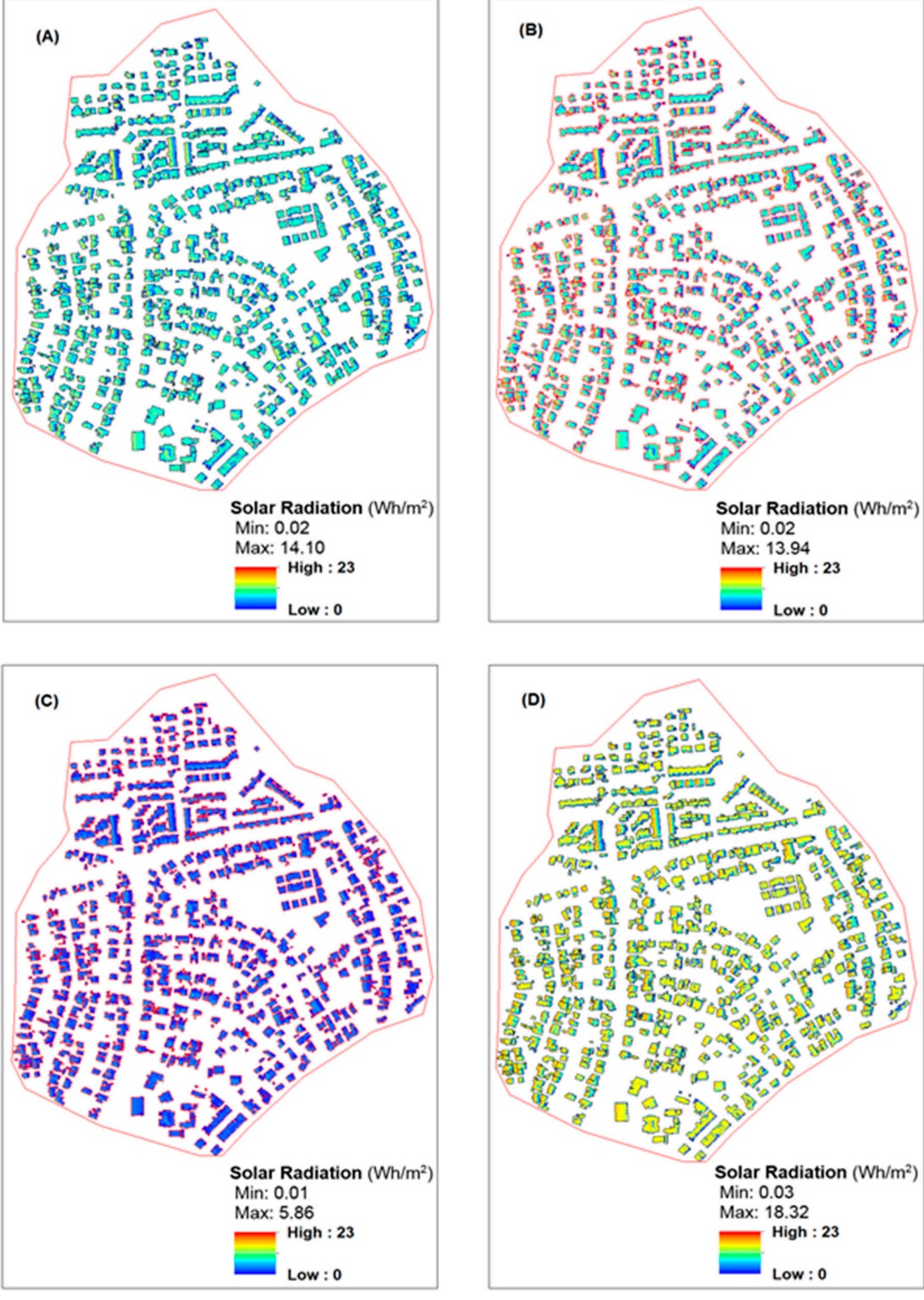

**Figure 12.** Incoming solar radiation (Wh/m$^2$) at 15:00: (**A**) vernal equinox (21 March), (**B**) autumnal equinox (23 September), (**C**) winter solstice (22 December), and (**D**) summer solstice (21 June).

*3.5. Spatial Analysis*

The total percentage cover of buildings in the study area is 25.6%. All building roofs (≥2 m) were considered for spatial analysis. After the applied method, a feature class with all related rooftop data (elevation, slope, aspect, and solar radiation) used to analyze maximum solar potential was extracted.

Figure 13 represents the total solar radiation distribution at a particular time (i.e., 9:00, 12:00, and 15:00) for four specific days of the year. For the equinox days (i.e., 21 March and 23 September), days and nights are equal, with "nearly" equal amount of daylight and darkness at all latitudes. Therefore, from the obtained analysis, it is seen that approximately the same solar insolation (2.03E+08 at 9:00, 3.05E+08 at 12:00, and 1.84E+08 at 15:00 for vernal equinox, 2.00E+08 at 9:00, 3.03E+08 at 12:00, and 1.81E+08 at 15:00 for autumnal equinox, in unit of KWh/m$^2$), with minor variation in radiation values, was achieved. In the northern hemisphere, on the winter solstice (22 December), days are shorter than nights. Hence, very low solar radiation is available (6.13E+07 at 9:00, 1.83E+08 at 12:00, and 5.68E+07 at 15:00 in unit of KWh/m$^2$), while on the summer solstice (21 June), days are longer and nights shorter, hence the highest solar radiation (2.92E+08 at 9:00, 3.79E+08 at 12:00, and 2.66E+08 at 15:00 in unit of KWh/m$^2$) is available as compared to the rest of the days. Figure 13 shows that the obtained solar radiation in the summer and winter solstices are highest and lowest, respectively. At 9:00, the aspect direction is east for all the days, and at 12:00, it is south for the equinox days and the winter solstice, while it is east for the summer solstice. At 15:00, the direction is west for both equinox days and the summer solstice, and south for the winter solstice. The better visualization of solar radiation distribution throughout the year can be well understood with the insolation pattern of these four specific days. It starts from low solar insolation (at the vernal equinox) to the highest (i.e., summer solstice), followed by a low value (i.e., autumnal equinox) to the lowest value (i.e., winter solstice), but the calculated values are the maximum values for a particular day of the year. The distribution of solar radiation with a slope-wise category for four specific days at a particular instant of time is shown in Figure 14. It can be seen that, for all the specified days, the gentle slope (4–22°) category gives the highest radiation at all times. In the study area, sun movement is from east to south to west (i.e., southern hemisphere). Therefore, the analysis suggests only three directions: east, west, and south. Overall, the analysis suggests that to obtain maximum solar radiation in the area, the solar panels must be installed with a 4–22° slope in east, west, and south-facing directions.

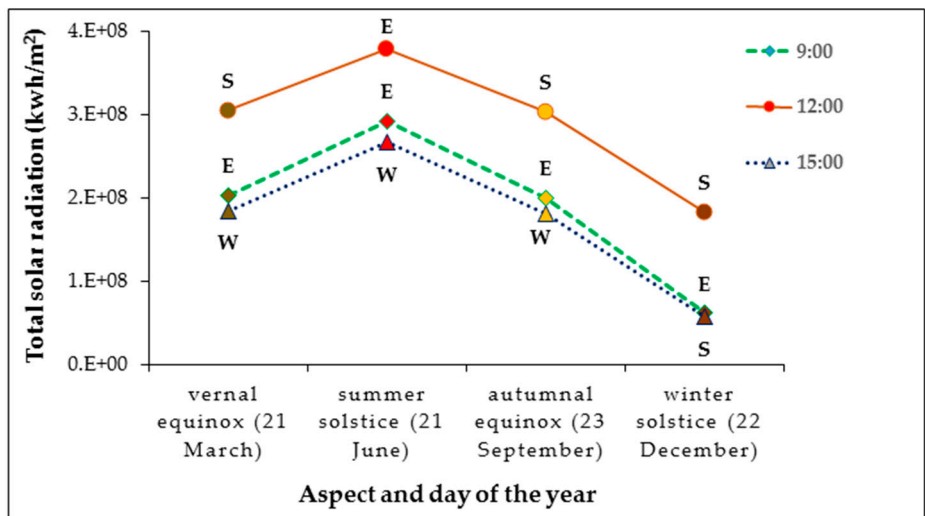

**Figure 13.** Total solar radiation distribution (including all slope categories), E (East), W (West), and S (South).

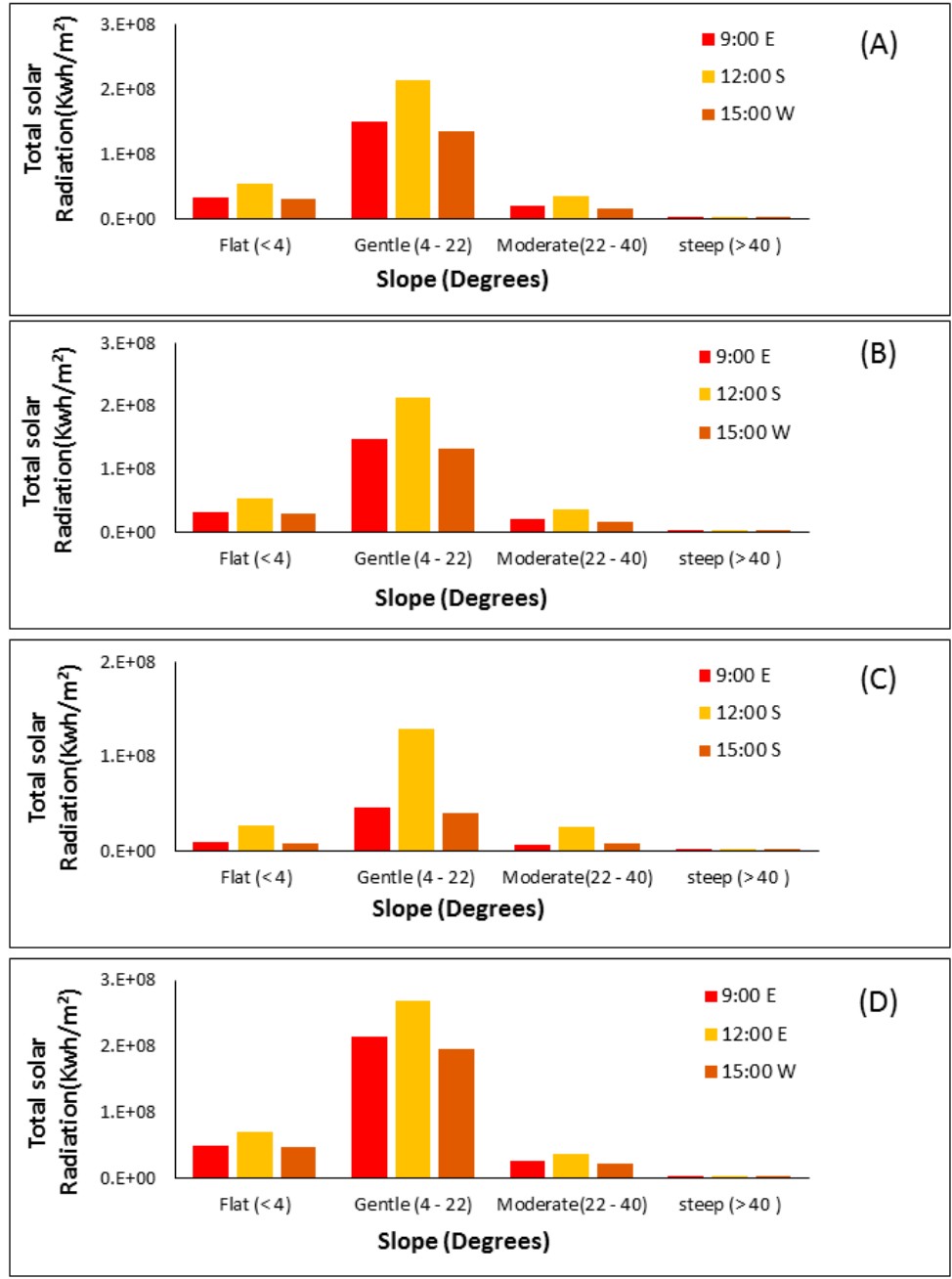

**Figure 14.** Total solar radiation (KWh/m$^2$) at specific times with slope category: (**A**) Vernal equinox (21 March), (**B**) autumnal equinox (23 September), (**C**) winter solstice (22 December), and (**D**) summer solstice (21 June).

## 4. Conclusions

To fulfill the energy demands of a growing population, the solar panels are a promising source. Therefore, it is necessary to install urban solar panels precisely in the appropriate positions to utilize maximum solar radiation effectively. For efficient solar power exploitation, the sun rays must be perpendicular to the solar panel. Therefore, accurate information about the slope and the aspect of the solar panel is required. This research study helps to find the aspect and slope of building rooftops that provide the maximum solar potential throughout the year. Previously, little research focused on building roofs using object-based procedures along with LiDAR data. Therefore, by capitalizing on established methods, this research has developed an object-based method, including orthophoto, which helps to classify and map building rooftops of various sizes and shapes, while airborne LiDAR

systems provide support to find the heights, slopes, and orientations of the corresponding building roof structures. The developed method enables the estimation of solar energy yields on flat and pitched roof surfaces, as well as multi-segment rooftops, through defining solar irradiances in units of pixels over a specific period.

The Kiryat Malakhi area in Israel was selected as the study area because of its infrastructure, comprising one to three floor buildings, with different roof types, flat and pitched, in diverse slopes and aspects along with multi-segment shapes. The area was analyzed based on the aspect-slope map. ESRI's Solar Analyst toolbox was used to produce solar maps for four specific days of the year and to measure the corresponding solar insolation output for individual grid cells. Finally, the instantaneous solar radiation was calculated at specific times, and the generated statistics are visualized graphically.

The results demonstrate that the applied method of calculation can achieve exact aspects and slopes at which to install solar panels on building roofs that will deliver high energy throughout the year, aiding in the effective utilization of renewable energy sources. The resultant solar insolation maps of specific days demonstrate the high energy throughout the year. Moreover, the generated aspect-slope map gives the optimal slope and direction for installing solar panels on building roofs that provide high solar energy.

The study used direct and diffuse solar radiation, not reflected radiation in the measurement. However, the reflected measurement could be used to obtain more precise insolation maps of building roofs (if the roof surface contains different types of materials). The lack of a cloud cover analysis is another key limitation of the current study. Still, the proposed method can be applied to more accurate solar panel sizing for installation and the precise calculation of solar radiation for tenants and commercial energy investors, resulting in the effective consumption of renewable energy sources. The developed algorithm can be applied to areas where rooftop information is required. In summary, the study can be expanded to estimate solar insolation on the facades of buildings, and future research can be undertaken for particular applications of solar devices.

**Author Contributions:** Conceptualization, A.T., A.K., I.A.M.; methodology, A.T.; software, A.T.; validation, A.T.; formal analysis, A.T.; writing: original draft preparation, A.T.; writing: review and editing, A.K., I.A.M.; supervision, A.K., I.A.M. All authors have read and agreed to the published version of the manuscript.

**Funding:** No funding was provided for this project.

**Acknowledgments:** We would like to thank Prabhakar Tripathi for his comments and suggestions.

**Conflicts of Interest:** The authors declare no conflict of interest.

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
