# Peer review of "Object-Based Image Procedures for Assessing the Solar Energy Photovoltaic Potential of Heterogeneous Rooftops Using Airborne LiDAR and Orthophoto"

_remotesensing, doi:10.3390/rs12020223_

Round 1

Reviewer 1 Report

Some suggestions:

Line 43. Solar energy dominates other renewable... in terms of what? For example, in terms of capacity, wind dominates the renewables. Explain clearly this statement.

Many abbreviations not explain clearly. E.g. ALK in line 74, DSM line 75, GRASS GIS line 78. Can you check the whole paper?

At the end of introduction, can you provide a paragraph to explain the structure of the paper? "Section 2 discuss .... Section 3 describes..."  

Equations 1-3, are they derived from [35]? it is not clear the source of these equations.

Line 241-242. Why is pitch roofs more difficult to visualise?

Tables should be typed, not included as 'image'/screen capture.

Line 296. 'Fu and Rich'. avoid symbol. Check the whole paper.

Fig 4 and 88. Font size in the flow chart is too big. Perhaps make the figure smaller.

Author Response

Thank you for taking the time to review this paper. Responses are below, inline.

Comment 1: Line 43. Solar energy dominates other renewable... in terms of what? For example, in terms of capacity, wind dominates the renewables. Explain clearly this statement.

Response 1:

The statement “Solar energy dominates other renewable and non-renewable energy resources [5]” has been rephrased as “(Among all the available renewable energy sources, the solar energy dominates over others in terms of its energy capacity growth rate and low maintenance cost [5,6])” and accordingly another reference has been added.

Comment 2: Many abbreviations not explain clearly. E.g. ALK in line 74, DSM line 75, GRASS GIS line 78. Can you check the whole paper?

Response 2: All the abbreviations have been corrected in the revised manuscript. Also several paragraphs of the “Introduction section” have been modified.

ALK map is now Trimble maps. Trimble is company that deals with the data of image processing as well as software.

Comment 3: At the end of introduction, can you provide a paragraph to explain the structure of the paper? "Section 2 discuss .... Section 3 describes..." 

Response 3: As per referee suggestion, in the “Introduction section” a paragraph was included in which the structure of the paper is defined properly.

Comment 4: Equations 1-3, are they derived from [35]? It is not clear the source of these equations.

Response 4: Only Equation 1 was derived from reference [35], while equation 2 and 3 were derived from applied object-based classification algorithm for more accurately separation of the buildings and vegetation classes.

Comment 5: Line 241-242. Why is pitch roofs more difficult to visualize?

Response 5: Here, visualization in terms of analysis. Pitched roof structures require a tedious mathematical analysis to identify exact slope of particular roof.

The word “visualize” seems confusing so replaced this with "analyze" in the revised manuscript.

Comment 6: Tables should be typed, not included as 'image'/screen capture.

Response 6: As the referee asked, all the tables in the revised manuscript were updated in MS word format.

Comment 7: Line 296. 'Fu & Rich' avoid and symbol. Check the whole paper.

Response 7: As per referee suggestion, a correction was done in revised manuscript.

Comment 8: Fig. 4 and 88. Font size in the flow chart is too big. Perhaps make the figure smaller.

Response 8: As suggested, font size and figure related issue has been corrected in the revised manuscript.

Reviewer 2 Report

The paper provides the estimation of solar energy deposition on building roofs. An object-method approach has been developed to assess the rooftop solar energy photovoltaic potential over an heterogenic urban environment, including surfaces at different inclinations and orientations, as well as multi-segment roofs in a single building. The purpose of the paper is very relevant and can be useful to optimize the use of renewable energy sources.

The paper is also well written, and certainly deserves to be publish in MDPI-Remote-sensing journal. Before that, I have simple and minor comments as follows, in order to ease the reading of paper.

Although they were not obtained at the same epoch, both LIDAR and orthophoto data were combined. Authors mention that it is assume “…. Minimal surface changes between 2012 and 2015.”. Fine with me. However, did authors confirmed this assumption ? Otherwise a few sentences on how the results could change are needed. Is the scale for distances in Figure 1 correct ? It seems to me it is not. In any case left scale (1A) is not coherent with the right scale (1B) Figures 10 (1-3), 11, 12: in these figures please use the correct format for the solar radiation unit. That is, Wh/m2. Not Wh/m2. The number 2 is an exponent. Same comment at many places within the manuscript. Figures 11 and 12: Use a more readable format for the solar radiation. For example, 2.5 108 and not 250000000.00 Accuracy of solar radiation measurements needs to be included in the text. Figure 12: solar radiation unit in the vertical axis (KWh/m2) is not coherent with that indicated in the caption (Wh/m2). The numbers in Figure 12 are in units of one of these 2, not both.

Author Response

Thank you for taking the time to review this paper. Responses are below, inline.

Comment 1: The paper provides the estimation of solar energy deposition on building roofs. An object-method approach has been developed to assess the rooftop solar energy photovoltaic potential over a heterogenic urban environment, including surfaces at different inclinations and orientations, as well as multi-segment roofs in a single building. The purpose of the paper is very relevant and can be useful to optimize the use of renewable energy sources.

The paper is also well written, and certainly deserves to be publish in MDPI-Remote-sensing journal. Before that, I have simple and minor comments as follows, in order to ease the reading of paper.

Although they were not obtained at the same epoch, both LIDAR and orthophoto data were combined. Authors mention that it is assume “…. Minimal surface changes between 2012 and 2015”. Fine with me. However, did authors confirmed this assumption? Otherwise a few sentences on how the results could change are needed. Is the scale for distances in Figure 1 correct? It seems to me it is not. In any case left scale (1A) is not coherent with the right scale (1B) Figures 10 (1-3), 11, 12: in these figures please use the correct format for the solar radiation unit. That is, Wh/m2. Not Wh/m2. The number 2 is an exponent. Same comment at many places within the manuscript. Figures 11 and 12: Use a more readable format for the solar radiation. For example, 2.5 108 and not 250000000.00 Accuracy of solar radiation measurements needs to be included in the text. Figure 12: solar radiation unit in the vertical axis (KWh/m2) is not coherent with that indicated in the caption (Wh/m2). The numbers in Figure 12 are in units of one of these 2, not both.

Response 1:

(A) The minimal surface changes between the mentioned dates were verified during the ground visual examination of the study area.  

(B) Scales in Figure 1(A) and Figure 1 (B) were updated in the revised manuscript.

(C) The solar radiation units in Figure 10-12, Figure 13 and Figure 14 were corrected in the revised manuscript.

(D) The resolutions of the figures 13 and 14 were approved so it becomes easily visualize and accuracies added in the text of the revised manuscript.

(E) Figure 14, vertical axis is coherent with caption now and updated in revised manuscript.

Reviewer 3 Report

Dear authors, the topic presented is very interesting. I think that the analysis is well conducted and it is potentially feasible to be published. However, a reviewer can focalize its attention for future studies. This paper can be improved, but I underline my positive impression.

the novelty of the paper is not well defined with existing and recent literature (please resolve this aspect in section 1 in a clear way). What is not present? How do you have intention to resolve it? Your methodology is based on some works? What are the differences? what is the limit? Results can be applied in other contexts? what is the comparison with other works? conclusions must be not redundanced what the future directions? literature is very weak. Please delete old works and examine recent works  i) https://onlinelibrary.wiley.com/doi/abs/10.1002/pip.2946 ii)https://www.mdpi.com/2076-0760/7/9/148 iii) https://www.sciencedirect.com/science/article/abs/pii/S036054421830834X you want publish in this journal but must be clarified also in literature section the quality of figures/tables in some cases is weak. there is a great difference with the style of the words in the text and the same in terms of dimensions. Please see guide for authors finally what is the different impact for several categories of actors in terms of value chain?

Author Response

Thank you for taking the time to review this paper. Responses are below, inline.

Comment 1: Dear authors, the topic presented is very interesting. I think that the analysis is well conducted and it is potentially feasible to be published. However, a reviewer can focalize its attention for future studies. This paper can be improved, but I underline my positive impression. The novelty of the paper is not well defined with existing and recent literature (please resolve this aspect in section 1 in a clear way). What is not present? How do you have intention to resolve it? Your methodology is based on some works? What are the differences? What is the limit? Results can be applied in other contexts? What is the comparison with other works? Conclusions must be not redundancy what the future directions? Literature is very weak. Please delete old works and examine recent works (i) https://onlinelibrary.wiley.com/doi/abs/10.1002/pip.2946 (ii)https://www.mdpi.com/2076-0760/7/9/148 (iii) https://www.sciencedirect.com/science/article/abs/pii/S036054421830834X you want publish in this journal but must be clarified also in literature section the quality of figures/tables in some cases is weak. There is a great difference with the style of the words in the text and the same in terms of dimensions. Please see guide for authors finally what is the different impact for several categories of actors in terms of value chain?

Response: 

The Introduction was modified in the revised manuscript, in consideration of reviewer's comments.

Few lines were removed due to the redundant nature: i.e., Line 49-54, Line 72-74, in the old version.

As per the referee suggestion, The Conclusion was updated to make it more precise in the revised manuscript.

Lines 608-617, in the old version, were removed to make conclusion more precise.

The comparison of the proposed studies cannot be done since this study covers whole area to estimate solar energy while previously reported studies mainly focused on specific variation of solar energy over a single building roof. The quality of figures/tables was improved for better visualization in the revised manuscript.

Reviewer 4 Report

Overall the paper has good analysis. Here are few of the comments:

Are all the tables created in the word document? If not, they must be redone in MS word for clarity. Figure 4 Caption must be reworded Figure 11 must be of high resolution Can the figures under the Appendix be added to the previous sections?  Conclusion must be made more precise.

Author Response

Thank you for taking the time to review this paper. Responses are below, inline.

Comment 1: Are all the tables created in the word document? If not, they must be redone in MS word for clarity. Figure 4 Caption must be reworded Figure 11 must be of high resolution Can the figures under the Appendix be added to the previous sections?  Conclusion must be made more precise.

Response 1:

(A) As per referee suggestion, all the tables were corrected in the revised manuscript.

(B) The caption of figure 4 was rewritten as “The implemented methodology to estimate PV potential".

(C) Figure 11 was updated to get a better visualization.

(D) As per the referee suggestion, the figures in appendix were moved to the previous section in the revised manuscript.

(E) As per the referee suggestion, the Conclusions section was updated to make it more precise in the revised manuscript.

Lines 608-617 were removed to make conclusion more precise.

Round 2

Reviewer 3 Report

Dear authors I don't see a significant change. Literature analysis and critical analysis of results must be' enhanced

Author Response

The authors would like to thank again the reviewer for spending his time in an attempt to improve our manuscript.  Since the general comment in the second round was related to the first round, we took these previous comments under serious consideration and replied, one by one. Responses are inline below.

Round-1

Comment 1: Dear authors, the topic presented is very interesting. I think that the analysis is well conducted and it is potentially feasible to be published. However, a reviewer can focalize its attention for future studies. This paper can be improved, but I underline my positive impression.

The novelty of the paper is not well defined with existing and recent literature (please resolve this aspect in section 1 in a clear way). What is not present? How do you have intention to resolve it? Your methodology is based on some works? What are the differences? What is the limit? Results can be applied in other contexts? What is the comparison with other works? Conclusions must be not redundancy what the future directions? Literature is very weak. Please delete old works and examine recent works (i) https://onlinelibrary.wiley.com/doi/abs/10.1002/pip.2946 (ii)https://www.mdpi.com/2076-0760/7/9/148 (iii) https://www.sciencedirect.com/science/article/abs/pii/S036054421830834X you want publish in this journal but must be clarified also in literature section the quality of figures/tables in some cases is weak. There is a great difference with the style of the words in the text and the same in terms of dimensions. Please see guide for authors finally what is the different impact for several categories of actors in terms of value chain?

The novelty of the paper is not well defined with existing and recent literature (please resolve this aspect in section 1 in a clear way). What is not present?

Response: - The novelty of the paper is related to a new methodological framework that ןד stressed in the lines 92-99 in the revised manuscript:

"Using the products of airborne LiDAR, a new model was developed to create an aspect-slope map. A minor change in slope and its direction over rooftops is noticeable while working on a single map (i.e., aspect-slope map) that could be overlooked while observing individual maps. Although various studies have been reported to estimate solar energy over building roofs but more specifically, to get the building footprints, the object-based method was applied to segment building as an object [33]. In some studies, building polygons was directly considered to estimate the PV potential [34–36]. To automate this tedious task, the present study develops an object-based algorithm to classify all buildings of the study area".

How do you have intention to resolve it?

Response: - To perform the analysis a very high-resolution data, such as ortho-rectified aerial photography (orthophotos), along with LiDAR data, was used and to accomplish the goal, data fusion techniques were developed.

Your methodology is based on some works? What are the differences? What is the limit?

Response: - Yes, the methodology is related to the previously reported studies in terms of estimating the PV potential on the building roofs (i.e., pixel-based) only. [Li, Y.; Ding, D.; Liu, C.; Wang, C. A pixel-based approach to estimation of solar energy potential on building roofs. Energy Build. 2016.] [Li, Y.; Liu, C. Estimating solar energy potentials on pitched roofs. Energy Build. 2017.].  

But the major differences are:

In the previously reported studies, at a time, only single type roof (flat roof and pitched roof) has been considered to estimate solar energy, while in the present study we have considered various roof types structure, i.e., flat and pitched, diverse slopes and aspects along with multi-segment shapes. In previously reported studies, building polygons and object-based methods were applied to segment building as an object. [33-36]. In preparation for building polygons "On-screen Digitization" process plays a vital role. The process is very tedious, time-consuming and it requires more precision to get an accurate vector file (building shapes). To resolve this issue and to automate this tedious task with more precision, this research study develops an object-based algorithm to classify all buildings of the study area. To estimate solar energy, the aspect and slope information plays a very crucial role [28,33–35]. In previously reported studies, individual aspect and slope maps were used to estimate solar energy. A minor change in slope and its direction over rooftops is not noticeable while observing individual maps. However, in the present research study, a new model was developed to create a single map (i.e., aspect-slope map) so that the overlooked minor changes can be observed.

Limits of the present research study:

In the study, only direct and diffuse solar radiation has been used to estimate solar energy. The study does not consider the cloud cover analysis. Results can be applied in other contexts? What is the comparison with other works?

Response: - This research study is useful to estimate solar potential that contains various roof types structure with diverse slopes and aspects along with multi-segment shapes. The comparison of the proposed studies cannot be done since this study covers the whole area to estimate solar energy while previously reported studies mainly focused on specific variations of solar energy over a single building roof.

Conclusions must be not redundancy what the future directions?

Response: - Redundancy has been removed from the conclusion section to make it more precise. Lines 608-617, in the old version, were removed to make the conclusion more precise.

Future directions of the research study:

The developed algorithm can be applied to areas where rooftop information is required; in summary, the study can be expanded to estimate solar insolation on the facades of buildings, and future research can be undertaken for particular applications of solar devices.

Literature is very weak. Please delete old works and examine recent works.

Response: -

Deleted old work in Literature:

In search of sustainable ecological development, significant efforts are invested in exploiting renewable energies and local governments offer an incentive for decreasing conventional energy expenses. Solar energy dominates other renewable and non-renewable energy resources [5].

(Reference: 5: - Buonomo, B.; Manca, O.; Montaniero, C.; Nardini, S. Numerical investigation of convective–radiative heat transfer in a building integrated solar chimney. Adv. Build. Energy Res. 2015.) 

The geospatial variety through a country involves various locations to assemble PV projects on building rooftops and distinctive motivating policies of governments [10]. The direct consumption of solar energy in buildings primarily occurs via solar PV technologies, the objective of which is to absorb solar radiation and produce electricity. Moreover, the exploitation of solar energy is very cost-effective. An investment in solar projects offers resilient energy for existing and future generations as compared to conventional energies. Renewable solar energy helps in reduced emission of GHGs that also support.

(Reference: 10:-Li, Y.; Si, T.; Liu, C. Geographical variation in energy yields of rooftop photovoltaic projects in Australia. Adv. Build. Energy Res. 2018.)

Agugiaro et al. [22] and Tereci et al. [23] created a digital surface model (DSM) using LiDAR data that was used to determine annual solar energy potential over recognized building roofs with the help of ALK map data and GIS software.

(References: 22:- Agugiaro, G.; Remondino, F.; Stevanato, G.; De Filippi, R.; Furlanello, C. Estimation of solar radiation on building roofs in mountainous areas. ISPRS - Int. Arch. Photogramm. Remote Sens. Spat. Inf. Sci. 2013.

23:- Tereci, A.; Schneider, D.; Kesten, D.; Strzalka, A.; Eicker, U. Energy saving potential and economical analysis of solar systems in the urban quarter scharnhauser park. In Proceedings of the 29th ISES Biennial Solar World Congress 2009, ISES 2009; 2009.)

Added recent work in Literature:

Among all the available renewable energy sources, solar energy dominates over others in terms of its energy capacity growth rate and low maintenance cost [5,6].

(References: 5:-IRENA Renewable Energy Now Accounts for a Third of Global Power Capacity. /newsroom/pressreleases/2019/Apr/Renewable-Energy-Now-Accounts-for-a-Third-of-Global-Power-Capacity 2019.

6:- IRENA Renewable capacity statistics 2019, International Renewable Energy Agency (IRENA), Abu Dhabi; 2019; ISBN ISBN 978-92-9260-123-2 (PDF)).

Without carbon cost [8]. The installation of photovoltaic panels on the building roofs has various advantages such as efficient utilization of renewable solar energy and distribution of total residential energy consumption, which leads to the reduction of CO2 emission [9].

(References: 8:- Solomon, A.A.; Bogdanov, D.; Breyer, C. Solar driven net zero emission electricity supply with negligible carbon cost: Israel as a case study for Sun Belt countries. Energy 2018, 155, 87–104.

9:-D’Adamo, I. The profitability of residential photovoltaic systems. A new scheme of subsidies based on the price of CO2 in a developed PV market. Soc. Sci. 2018, 7, 1–21.)

The previously reported studies show that the aspect and slope information plays a very crucial role in solar estimation analysis [33–35].

(References: 33:- Huang, Y.; Chen, Z.; Wu, B.; Chen, L.; Mao, W.; Zhao, F.; Wu, J.; Wu, J.; Yu, B. Estimating roof solar energy potential in the downtown area using a GPU-accelerated solar radiation model and airborne LiDAR data. Remote Sens. 2015, 7, 17212–17233.

Bayrakci Boz, M.; Calvert, K.; R. S. Brownson, J. An automated model for rooftop PV systems assessment in ArcGIS using LIDAR. AIMS Energy 2015, 3, 401–420. Wong, M.S.; Zhu, R.; Liu, Z.; Lu, L.; Peng, J.; Tang, Z.; Lo, C.H.; Chan, W.K. Estimation of Hong Kong’s solar energy potential using GIS and remote sensing technologies. Renew. Energy 2016, 99, 325–335.)

 3. Although various studies have been reported to estimate solar energy over building roofs but more specifically, to get the building footprints, the object-based method was applied to segment building as an object [33]. In some studies, building polygons was directly considered to estimate the PV potential [34-36].

(Reference: 36:- Suomalainen, K.; Wang, V.; Sharp, B. Rooftop solar potential based on LiDAR data: Bottom-up assessment at neighbourhood level. Renew. Energy 2017, 111, 463–475.)

The quality of figures/tables in some cases is weak. There is a great difference with the style of the words in the text and the same in terms of dimensions.

Response: - The quality of figures/tables was improved for better visualization in the revised manuscript.

Round-2

Comment 1: Dear authors I don't see a significant change. Literature analysis must be' enhanced.

Response: - All the corrections and added new references in the "Introduction section" were highlighted in the responses mentioned above.

Comment 2: Critical analysis of results must be' enhanced.

Response: - In the subsection "3.4 Spatio-Temporal Distribution of Solar Radiation" the following statements were added to enhance the description of Figures 10-12.  

In the generated solar radiation maps, the solar radiation values were scaled from "0 to 23" for better visualization of the differences between the specific days of the year. Figures 10-12 reveal that the winter solstice (22 December) and summer solstice (21 June) received the minimum and maximum solar radiation, respectively. The Equinox days, however, have minor changes in radiation values.